# Mechanistic Investigation of WWOX Function in NF-kB-Induced Skin Inflammation in Psoriasis

**DOI:** 10.3390/ijms25010167

**Published:** 2023-12-21

**Authors:** Min-Jeong Shin, Hyun-Sun Kim, Pyeongan Lee, Na-Gyeong Yang, Jae-Yun Kim, Yun-Su Eun, Whiin Lee, Doyeon Kim, Young Lee, Kyung-Eun Jung, Dongkyun Hong, Jung-Min Shin, Sul-Hee Lee, Sung-Yul Lee, Chang-Deok Kim, Jung-Eun Kim

**Affiliations:** 1Department of Dermatology, College of Medicine, Soonchunhyang University Cheonan Hospital, Cheonan 31151, Republic of Koreadmyang2@naver.com (N.-G.Y.);; 2Department of Dermatology, Soonchunhyang University Graduate School of Medicine, Asan 31538, Republic of Koreapyeongani13@gmail.com (P.L.); 3Department of Dermatology, School of Medicine, Chungnam National University Hospital, Daejeon 35015, Republic of Korearesina20@gmail.com (Y.L.);; 4Department of Dermatology, College of Medicine, Soonchunhyang University Bucheon Hospital, Bucheon 14584, Republic of Korea; 5Department of Medical Science, School of Medicine, Chungnam National University, Daejeon 35015, Republic of Korea

**Keywords:** psoriasis, WWOX, poly(I:C), NF-kB, PKC, calcium ion

## Abstract

Psoriasis is a chronic inflammatory skin disease characterized by epidermal hyperproliferation, aberrant differentiation of keratinocytes, and dysregulated immune responses. WW domain-containing oxidoreductase (WWOX) is a non-classical tumor suppressor gene that regulates multiple cellular processes, including proliferation, apoptosis, and migration. This study aimed to explore the possible role of WWOX in the pathogenesis of psoriasis. Immunohistochemical analysis showed that the expression of WWOX was increased in epidermal keratinocytes of both human psoriatic lesions and imiquimod-induced mice psoriatic model. Immortalized human epidermal keratinocytes were transduced with a recombinant adenovirus expressing microRNA specific for WWOX to downregulate its expression. Inflammatory responses were detected using Western blotting, real-time quantitative reverse transcription polymerase chain reaction (PCR), and enzyme-linked immunosorbent assay. In human epidermal keratinocytes, WWOX knockdown reduced nuclear factor-kappa B signaling and levels of proinflammatory cytokines induced by polyinosinic: polycytidylic acid [(poly(I:C)] in vitro. Furthermore, calcium chelator and protein kinase C (PKC) inhibitors significantly reduced poly(I:C)-induced inflammatory reactions. WWOX plays a role in the inflammatory reaction of epidermal keratinocytes by regulating calcium and PKC signaling. Targeting WWOX could be a novel therapeutic approach for psoriasis in the future.

## 1. Introduction

Psoriasis is a chronic, immune-mediated inflammatory skin disease that affects approximately 2% of the general population. It is characterized by well-demarcated, erythematous plaques with silvery scales [1,2]. The disease imposes a significant burden due to its numerous comorbidities, including psoriatic arthritis and cardiovascular diseases, and detrimentally impacts patients’ social and emotional quality of life. Although its etiology and pathogenesis remain unclear, many studies have suggested that various environmental and genetic triggers cause immune system dysregulation and a cascade of inflammatory reactions that eventually lead to psoriasis. The histologic hallmark of psoriasis involves epidermal hyperplasia with dermal inflammatory cell infiltrates composed of primarily dermal dendritic cells, T lymphocytes, neutrophils, and macrophages. Proinflammatory cytokines, including tissue necrosis factor-alpha (TNF-α), interferon-gamma (INF-γ), interleukin-1 beta (IL-1β), IL-17, and IL-23 produced by activated inflammatory cells elicit hyperproliferation and impaired differentiation of keratinocytes. These inflammatory events ultimately result in a characteristic psoriatic phenotype [3,4,5,6].

Advancements in understanding the pathogenesis of psoriasis have led to the development of highly effective and safe targeted therapeutic agents. These novel biologics, which specifically target molecules such as TNF-α, IL-17, and IL-23, have enabled revolutionary improvements in treating severe psoriasis. Current therapeutic strategies primarily focus on disrupting IL-17 or IL-23 cytokine signaling. IL-17 inhibitors, including secukinumab, ixekizumab, and brodalumab, along with IL-23 inhibitors, including guselkumab, risankizumab, and tildrakizumab, represent the latest class of biologics available for the management of psoriasis, bringing the goal of “complete clear skin” closer to a realistic goal [7,8,9]. However, patients with mild to moderate psoriasis, comprising the majority, are challenged to use biologic agents. In addition, there are refractory areas where biologics may not provide sufficient effects, and some patients might not respond well to biologics. Therefore, further research on the mechanism of recalcitrant psoriasis and its new treatment remains necessary. While biologics have shown remarkable therapeutic benefits in psoriasis, certain limitations, such as subcutaneous injection and their high cost, lead to interest in other treatment options, such as small-molecule drugs called Janus kinase (JAK) inhibitors. The IL-23 receptor depends on a heterodimer of JAK2 and TYK2 for signal transduction. Therefore, JAK inhibitors have the potential to offer an oral alternative to the currently available systemic therapeutics, including methotrexate, apremilast, and biologics [10]. Deucravacitinib is the first oral selective tyrosine kinase 2 (TYK2) inhibitor responsible for the signaling of IL-23 and other cytokines involved in the pathogenesis of psoriasis [11,12]. JAK 1/3 inhibitor tofacitinib and the selective JAK1 inhibitor upadacitinib have been also approved in psoriatic arthritis [13].

The human WW domain-containing oxidoreductase (WWOX) gene is located with the fragile site FRA16D at 16q23.3–24.1 [14]. WWOX is widely recognized as a tumor suppressor, and a substantial portion of WWOX functional investigations has elucidated its role in modulating cancer related pathways through interactions between proteins [15]. Furthermore, it interacts with various transcription factors and signaling molecules that contain proline-tyrosine-rich motifs at established crossroads between inflammation and cancer [16]. The decrease in WWOX expression, observed in diverse cancer types including lung and ovarian cancer, correlates with increased tumor metastasis and poorer prognosis. However, the specific mechanisms underlying this connection remain largely unexplored [17]. WWOX has been shown to play a crucial role not only in cancer development but also in various other cellular processes [18]. Several studies indicate WWOX involvement in high-density lipoprotein (HDL) cholesterol, triglyceride, and other lipoprotein metabolism [19,20]. Additionally, WWOX deficiency can lead to decreased mitochondrial oxidation, increased glycolysis, and fatty acid oxidation, ultimately contributing to the development of metabolic syndrome. Further research demonstrated a significant association between WWOX variants and diabetes [21,22]. Dysregulated metabolic states might contribute to the expression of inflammatory diseases by inducing inflammation and oxidative stress. Recently, WWOX has also been shown to act on TGF-B overexpression, a key mediator in the relationship between intestinal inflammation and fibrosis in inflammatory bowel disease [16]. Furthermore, a novel IkBα/WWOX/ERK signal pathway that stimulates T cell maturation has been identified. In addition to the calcium ionophore A23187 and phorbol ester, human ALL MOLT-4 cells undergo terminal maturation toward a T cell phenotype, a process mediated by IkBα/WWOX/ERK signaling [23,24]. Psoriasis is recognized as an immune-mediated skin disorder closely linked to T cell activity. Recent research has highlighted the significance of the ERK signaling pathway as a crucial target for managing angiogenesis and understanding the underlying mechanisms of psoriasis [25]. In this context, a potential connection between WWOX and psoriasis is assumed. In a study involving the global DNA methylation profiling of psoriatic skin, the WWOX gene was found to be strongly hypomethylated and exhibited increased transcription levels in psoriatic skin compared to normal skin [26,27]. However, there is still a lack of mechanistic investigation regarding WWOX and its association with psoriasis.

Extracellular calcium is critical in epidermal barrier function and keratinocyte differentiation [28], and is also involved in the inflammatory response. The synthesis of calcium-sensing receptors (calcium regulators) is upregulated by proinflammatory cytokines such as IL-6 and IL-1β. Intracellular calcium accumulates, and its release can trigger inflammation [29,30]. Activation of conventional protein kinase C (PKC) enzymes requires calcium, and activated PKC can then influence the canonical nuclear factor-kappa B (NF-kB) signaling pathway [31,32]. The NF-kB transcription factor is active in many cell lines and is a crucial regulator of inflammation and other complex biological processes. Activated NF-kB levels are significantly elevated in the lesional skin of patients with psoriasis [33,34]. This leads to the transcription of various inflammatory signals, such as proinflammatory cytokines (TNF-α, IL-1, IL-6), chemokines, and Toll-like receptor ligands, which can contribute to the inflammatory process in psoriasis. Dysregulation of NF-κB signaling can alter keratinocyte and immune responses through its effect on cellular proliferation, differentiation, and apoptosis, as well as cytokine production and recruitment. These alterations result in the characteristic thickening and scaling of the skin in psoriatic lesions. Treatments for psoriasis have also been developed to inhibitor NF-κB activation or downstream transcription factors, including TNF-α or IL-17/IL-23 [35].

This study aimed to investigate the association of WWOX with psoriatic inflammation. A better understanding of the effect of WWOX associated with PKC signaling on psoriasis might provide novel insights into the mechanisms underlying the development of psoriasis.

## 2. Results

### 2.1. WWOX Is Highly Expressed in Psoriatic Skin

To investigate the relationship between WWOX and psoriasis, we first performed IHC against WWOX using skin specimens obtained from normal and psoriatic lesions. Weak WWOX immunoreactivity was detected in the normal epidermis, whereas epidermal keratinocytes of psoriatic lesions exhibited strong WWOX immunoreactivity (Figure 1A). A well-established mouse model of IMQ-induced psoriasis was used to determine whether psoriasis was associated with WWOX expression. The timeline of the mouse model study, along with representative images and Hematoxylin and Eosin (H&E) staining findings, is presented in Appendix A. The epidermal thickness was markedly increased when the IMQ cream was applied topically to the mouse’s back skin. Like human psoriatic lesions, WWOX immunoreactivity was strong in the thickened epidermis of IMQ-treated mouse skin (Figure 1B). WWOX expression was upregulated in both human psoriatic lesions and the IMQ-induced mouse psoriasis model. Additionally, we analyzed public data on WWOX expression in lesional and non-lesional skin biopsies of psoriasis using Gene Expression Omnibus (GEO) profiles provided by NCBI. In these data, WWOX expression was higher in lesional than non-lesional skin, consistent with our results (Appendix A) [36].

### 2.2. WWOX Knockdown Inhibits Poly(I:C)-Induced Inflammatory Response in Epidermal Keratinocytes

Epidermal keratinocytes express Toll-like receptor 3 (TLR3). Expression of TLR3 in epidermal keratinocytes plays an essential role in innate immune-related inflammatory reactions [37]. Poly(I:C), a double-stranded RNA (dsRNA) analog, recognizes TLR3 and induces an inflammatory response. It is widely used in a study on diseases related to the innate immune response, such as psoriasis [38,39]. WWOX expression was controlled to investigate poly(I:C)-induced inflammatory response associated with WWOX using miR-scr and miR-WWOX in the present study. SV-keratinocytes were treated with 0.5 μg/mL poly(I:C) for 1 hr. Results revealed that WWOX expression in the miR-WWOX-treated group was reduced compared to that in the miR-scr-treated group (Figure 2A). Treatment with poly(I:C) increased the protein levels of the NF-kB marker phospho-p65 in keratinocytes. After the adenovirus-mediated knockdown of WWOX expression in these cells, the protein levels of the NF-kB marker phospho-p65 were decreased (Figure 2B). Original uncropped scans of the blots are presented in Appendix A. The rescue experiment showed the over-expression of WWOX increased p-p65, which was decreased by WWOX knockdown (Appendix A).

To evaluate whether WWOX knockdown affected the expression of proinflammatory cytokines, we performed qRT-PCR and ELISA. WWOX gene knockdown in keratinocytes was verified by qRT-PCR (Figure 3A). SV-keratinocytes were treated with 0.5 μg/mL poly(I:C) for 2 h. Treatment with poly(I:C) induced a variety of mRNAs of proinflammatory cytokines, including IL-6, IL-8, TNF-α, IL-1β, and Chemokine ligand 20. WWOX knockdown significantly decreased levels of mRNAs encoding these proinflammatory cytokines and their secretion (Figure 3B,C). These results suggest that WWOX knockdown can suppress proinflammatory reactions in keratinocytes.

### 2.3. PKC Activity Is Decreased by WWOX Knockdown in Poly(I:C)-Treated Group

We performed QIAGEN Ingenuity Pathway Analysis (IPA) to find WWOX-interacting proteins, and the results predicted that WWOX could interact with JPT2 (Appendix A). JPT2 is known to regulate intracellular calcium levels by activating the ryanodine receptor in the endoplasmic reticulum [40]. It is well known that protein kinase C (PKC) activity is regulated by changes in intracellular calcium levels in keratinocytes, so we assumed that WWOX could affect PKC signaling. First, we treated epidermal keratinocytes with poly(I:C) to determine whether poly(I:C) enhances PKC activity. Samples were collected over time. Initially, PKC activity gradually increased, peaking at 1 h and then decreasing after 2 h (Figure 4A). When cells expressing miR-WWOX were treated with poly(I:C) for 1 h, the PKC activity decreased under the WWOX knockdown condition (Figure 4B), suggesting that WWOX knockdown could regulate PKC activity.

### 2.4. Inflammatory Response by Ca^2+^ and PKC

Calcium is involved in inflammatory diseases [41,42]. However, whether calcium ion plays a role in the inflammatory responses of epidermal keratinocytes remains unclear. Therefore, inflammatory responses of epithermal keratinocytes in the presence of BAPTA-AM, a calcium chelator, were assessed. SV-keratinocytes were pre-treated with BAPTA-AM for 1 h and then treated with 0.5 μg/mL poly(I:C) for 2 h. In the poly(I:C)-treated group, BAPTA-AM reduced levels and secretion of proinflammatory cytokines in a dose-dependent manner (Figure 5A,B). GF109203X, a PKC inhibitor, decreased mRNA levels and secretion of proinflammatory cytokines dose-dependently. SV-keratinocytes were pre-treated with GF109203X for 1 h and then treated with 0.5 μg/mL poly(I:C) for 2 h. GF109203X reduced the mRNA expression of IL-6 and IL-1β significantly. Also, the mRNA expression of IL-8 and CCL20 decreased but was not statistically significant. The secretion levels of IL-6, IL-8, IL-1β, and CCL20 were decreased gradually by dose, which was statistically significant (Figure 5C,D). It also significantly reduced the expression of p-P65 (Figure 5E), a marker of NF-kB, suggesting that calcium and PKC could mediate skin inflammatory responses by increasing levels of proinflammatory cytokines. Original uncropped scans of the blots are presented in Appendix A.

## 3. Discussion

WWOX, located within the chromosomal region 16q23.3–24.1, was initially recognized as a gene commonly subject to loss of heterozygosity in a variety of cancer types [43,44,45]. The gene spans 1.1 million bases and comprises 9 exons, with intron 9 housing the common fragile site FRA16D, which significantly contributes to instability in the 16q23.3–24.1 region. The localization of WWOX within these unstable chromosomal regions was the initial evidence pointing toward its role as a tumor suppressor gene [44,46]. Previously known as a proapoptotic tumor suppressor, WWOX has been reported to be involved in epidermal homeostasis in normal human skin. The absence of WWOX in both humans and animals results in growth retardation and premature death during postnatal development. Skin integrity is crucial for the survival of organisms as it acts as a barrier against various environmental factors. One study showed that WWOX protein is expressed in human epidermal suprabasal cells, and its expression progressively intensifies as the cells undergo differentiation towards the superficial layers [47]. WWOX is also involved in various biological processes, including bubbling apoptosis, transcription, RNA processing, and genomic stability, suggesting that WWOX plays a critical role in normal cellular/physiological homeostasis maintenance [48]. Recent studies have indicated that WWOX plays a relevant role in the inflammatory response, and an association between WWOX and the NF-kB signaling pathway has also been demonstrated [49,50].

Psoriasis is a chronic and immunologically mediated inflammatory disorder. While its prevalence varies among populations, it is known to affect approximately 2 to 3% of the population worldwide. Research on the complex molecular mechanisms of psoriasis is constantly in progress. A significant hallmark in the pathophysiology of psoriasis involves epidermal hyperproliferation, coupled with the infiltration of inflammatory cells into both the epidermis and dermis. The most prevalent form, psoriasis vulgaris, is characterized by erythematous plaques often covered by a thick layer of silvery scales. Key contributors to the pathogenesis of psoriasis encompass keratinocytes, plasmacytoid dendritic cells, macrophages, and fibroblasts. These cells release specific inflammatory mediators, such as tumor necrosis factor-alpha, IL-1, and IL-6, which subsequently drive keratinocyte proliferation. The release of these inflammatory mediators activates myeloid dendritic cells, prompting the secretion of IL-12 and IL-23 by naïve T cells, thus facilitating the differentiation of naïve T cells into T helper (Th)1 and Th17 cells [51,52,53]. A recent study highlighted correlations between psoriasis severity and TNF-α-related mitogen-activated protein kinase, NF-kB, and Janus kinase pathways [54]. NF-kB, a pivotal protein transcription factor, coordinates complex biological processes primarily centered around inflammation. Given its regulatory influence across diverse immune and inflammatory pathways, as well as in cell proliferation, differentiation, and apoptosis, NF-kB is postulated to have a significant role in the pathogenesis of psoriasis [35]. Lesional samples of psoriatic skin exhibit increased levels of activated and phosphorylated NF-kB in comparison to non-psoriatic skin. Sequential biopsies obtained from psoriasis patients undergoing treatment with etanercept, a TNF antagonist used for psoriasis therapy, demonstrated a significant reduction in phosphorylated NF-kB/RelA [34].

We hypothesized that WWOX could be involved in keratinocytes and their inflammatory responses, which play an essential role in the pathogenesis of psoriasis. In this study, we investigated the correlation of WWOX with poly(I:C)-induced NF-kB signaling pathway in epidermal keratinocytes, which is thought to be critical to the pathogenesis of psoriasis. Our research showed increased expression of WWOX in the epidermis of psoriasis patients and the IMQ-treated mouse model. Furthermore, we observed that WWOX knockdown reduced levels of poly(I:C)-induced proinflammatory cytokines and NF-kB marker p-P65. The synthetic viral dsRNA analog poly(I:C) could induce an innate immune response via TLR3 activation. Keratinocytes, by detecting danger signals by the TLR system, initiate innate immune responses. Among various TLRs expressed by keratinocytes, TLR3 can recognize dsRNA synthesized during the replication of some viruses. Psoriatic keratinocytes are susceptible to viral dsRNA and, upon recognition, could produce proinflammatory cytokines associated with psoriasis.

Additionally, NF-kB is a key transcription factor responsible for the pathogenesis of psoriasis. Activation of NF-kB can stimulate the production of proinflammatory cytokines. Various inflammatory cytokines such as IL-1β, IL-6, and TNF-α produced by immune cells contribute to the proliferation of keratinocytes [55]. This suggests a potential role for WWOX in regulating poly(I:C)-induced TLR3-mediated inflammation of epidermal keratinocytes through an NF-kB signaling process.

Calcium is a significant regulator of keratinocyte differentiation, skin barrier permeability, and homeostasis [56]. Changes in calcium levels of keratinocytes can trigger psoriasis, skin cancer, and disturb the skin barrier [57]. The PKC family of protein kinases is known to control the functions of other proteins via the phosphorylation of hydroxyl groups of serine and threonine residues. In turn, PKC enzymes are activated by increased diacylglycerol or calcium concentrations [58]. PKC plays a fundamental role in the immune system by regulating signaling pathways important for innate and adaptive immunity [59]. Along with immune cells, including T lymphocytes, PKC expressed in keratinocytes is involved in keratinocyte differentiation and skin inflammation, thus contributing to the development of psoriasis. Transgenic mice overexpressing PKCα showed skin barrier disruption and severe intraepidermal neutrophilic inflammation [60,61].

Moreover, PKCs can modulate NF-kB activity, essential in skin inflammation [62]. As PKCs play critical roles in keratinocyte differentiation, inflammatory response, and activation of NF-kB signaling, we expected that PKCs would be involved in WWOX-mediated regulation of epidermal keratinocyte inflammation. Thus, our investigation focused on examining PKC activity subsequent to WWOX knockdown and treatment with poly(I:C). We also explored the inflammatory response in the presence of calcium chelator and PKC inhibitor. Our findings confirmed that the PKC activity was increased by poly(I:C). In addition, calcium chelator or PKC inhibitor reduced levels of proinflammatory cytokines in a dose-dependent manner.

Interestingly, upon WWOX knockdown, there was no change in PKC activity in the group not treated with poly(I:C), whereas PKC activity was decreased in the group treated with poly(I:C). The outcome of this study demonstrated the involvement of calcium and PKC in the inflammatory response of epidermal keratinocytes and the influence of WWOX knockdown on their levels. These findings support the hypothesis that WWOX can modulate the pathogenesis of inflammatory skin diseases such as psoriasis.

This study, while providing new perspectives on the role of WWOX in psoriasis, has several limitations that warrant discussion. The primary focus lies on an in vitro model employing poly(I:C) to mimic NF-kB inflammation, a simulation that might not fully encapsulate the complexities inherent in human and mouse models of psoriasis. Notably, WWOX expression was observed to increase following poly(I:C) treatment, a change that was statistically significant, especially in the context of WWOX knockdown. However, our study is predominantly centered on the NF-kB signaling pathway and its modulation by WWOX, potentially overlooking other critical pathways implicated in the pathogenesis of psoriasis. Furthermore, in our attempts to elucidate the role of WWOX in this context, we encountered a limitation in not being able to fully restore p-p65 levels to those observed prior to WWOX knockdown, even after a successful rescue experiment. This finding suggests a complex interplay of factors in the WWOX-mediated modulation of NF-kB, necessitating more detailed exploration. Additionally, the roles of calcium and PKC in the inflammatory response of epidermal keratinocytes, as revealed in our study, call for further investigation to more precisely define their exact roles in psoriasis. Our use of PKC inhibitors and calcium chelators to modulate inflammation highlights potential therapeutic strategies, but these findings are preliminary and need to be validated through more comprehensive in vivo studies. Considering the limited scope of our current research models, further studies to verify the function of WWOX in various in vitro models of psoriatic inflammation will be needed in the future.

In summary, we demonstrated that WWOX was increased in the epidermis of psoriasis and that WWOX knockdown reduced levels of proinflammatory cytokines, the NF-kB marker p-P65. Most importantly, data presented here suggest that WWOX might modulate the NF-kB pathway triggered by PKC activation in epidermal keratinocytes which indicates that the involvement of WWOX could be a critical molecular event for the pathogenesis of psoriasis and that targeting WWOX could be a novel therapeutic approach for psoriasis in the future. Further research is needed to understand the mechanisms of WWOX in immune-mediated skin diseases, including psoriasis.

## 4. Materials and Methods

### 4.1. Immunohistochemistry (IHC)

For the immunohistochemical analysis, the skin tissues were fixed using a solution of 10% formaldehyde (*v*/*v*). Subsequently, these tissues were embedded in paraffin, sectioned into slices measuring 4 μM-thick sections, deparaffinized with xylene, and then rehydrated gradually with a series of alcohol baths. In preparation for the immunohistochemical staining procedure, the sections underwent treatment with 3% (*v*/*v*) H_2_O_2_ solution to block endogenous peroxidase. Following this, the sections were gently incubated with an IHC protein block solution (DAKO, Carpinteria, CA, USA, S2002). Once properly conditioned, the sections were subjected to an overnight incubation at a temperature of 4 °C with a primary WWOX antibody (Abcam, Cambridge, UK, ab238144). Subsequently, they were then incubated with horseradish peroxidase-conjugated secondary antibodies (DAKO, K4003) for 2 h at room temperature. After washing, the sections were exposed to a solution of diaminobenzidine tetrahydrochloride and counterstained with Mayer’s hematoxylin. Quantitative analysis of immunohistochemical staining was performed using ImageJ software v1.53a (WS Rasband, National Institutes of Health, Bethesda, MD, USA).

### 4.2. Imiquimod (IMQ)-Induced Psoriasis Mouse Model

To establish the IMQ-induced psoriasis model, we selected female BALB/c mice obtained from Orient Bio (Seongnam, Republic of Korea), each aged seven weeks, as described previously [63]. The mice underwent depilation of their back hair through the utilization of electric clippers. Following this, Aldara^TM^ cream (6.25 µg, IMQ 5%, Dong-A ST, Seoul, Republic of Korea) was topically applied for 14 consecutive days.

### 4.3. Cell Culture and Reagents

Human skin tissues were obtained from Chungnam National University Hospital (Daejeon, Republic of Korea). The Institutional Review Board approved all procedures of Chungnam National University Hospital (IRB No.1011-135). Informed written consent was obtained from each donor. Primary keratinocytes were isolated from the epidermis, immortalized using a recombinant retrovirus expressing simian virus 40 T antigen [64], and cultured at 37 °C with 5% carbon dioxide in keratinocyte-serum free medium with a bovine pituitary extract and recombinant human epidermal growth factor (Thermo Scientific, Rockford, IL, USA, PHG0311). SV-keratinocytes were treated with 0.5 μg/mL poly(I:C) (Invitrogen, Carlsbad, CA, USA, vac-pic) for 1 h or 2 h. BAPTA-AM (Selleck Chemicals, Houston, TX, USA, S7534) and GF109203X (Selleck Chemicals, S7534) were dissolved in dimethyl sulfoxide at 60 min before adding poly(I:C).

### 4.4. Recombinant Adenovirus Expressing a WWOX-Specific microRNA and WWOX Knockdown in Human Keratinocytes

The microRNA sequence to target human WWOX messenger RNA (mRNA) was obtained with an RNAi Designer online tool (Invitrogen, Carlsbad, CA, USA). The double-stranded DNA oligonucleotides were synthesized and cloned into the parental vector pcDNA6.2-GW/EmGFP-miR. The expression cassette for microRNA was inserted into the pENT/CMV vector, and then the adenovirus was prepared. The synthesized sequences were as follows: top strand 5′-TGCTGTTATCATCCACAGTAAAC GCCGTTTTGGCCACTGACTGACGGCGTTTAGTGGATGATAA, bottom strand 5′-CC TGTTATCATCCACTAAACGCCGTCAGTCAGTGGCCAAAACGGCGTTTACTGTGATGATAAC. The resulting microRNA sequence targets nt 395~415 in NM_016373.4. Human epidermal keratinocytes were transduced with an adenovirus expressing scrambled microRNA (Ad/miScr) and an adenovirus expressing WWOX-specific microRNA (Ad/miWWOX) at a multiplicity of infection of 10 for 6 h [65].

### 4.5. Western Blot Analysis

Cells were harvested using a centrifuge and lysed in PRO-PREP solution (iNtRON Biotechnology, Seongnam, Republic of Korea, 17081). Total protein levels were quantified using a bicinchoninic acid protein assay kit (Thermo Scientific, Rockford, IL, USA, 23225). Protein samples were subjected to sodium dodecyl sulfate-polyacrylamide gel electrophoresis and transferred to nitrocellulose membranes (Pall Corporation, Port Washington, NY, USA, 66485). After blocking with 5% (*w*/*v*) skim milk (MB cell, Seoul, Republic of Korea, MB-S1667), membranes were incubated with primary antibodies at 4 °C overnight. After washing off unbound primary antibodies, membranes were incubated with peroxidase-conjugated secondary antibodies. Protein bands were visualized using an enhanced chemiluminescence system (TransLab, Daejeon, Republic of Korea, TLP-112.1) after incubation at room temperature for 2 h. The following primary antibodies were used in this study: anti-WWOX (Abcam, ab28144), anti-phospho-NF-kB p65 (Cell Signaling Technology, Danvers, MA, USA, #3033), and anti-actin (Santa Cruz Biotechnology, Santa Cruz, CA, USA, sc-47778). Protein bands were quantified by ImageJ software v1.53a (National Institutes of Health).

### 4.6. Quantitative Real-Time Polymerase Chain Reaction

Total RNA was isolated from cultured epidermal keratinocytes using an Easy-blue RNA extraction kit (iNtRON Biotechnology, 17061). Total RNA (2 μg) was reverse transcribed to complementary DNA with an M-MLV reverse transcriptase. The resulting complementary DNA was amplified using a SYBR Green Master Mix kit (ELPIS Biotech, Daejeon, Republic of Korea, EBT-1802). Real-time quantitative reverse transcription polymerase chain reaction (qRT-PCR) was performed using an AB StepOne Real-Time PCR system (Applied Biosystems, Foster City, CA, USA). The following gene-specific primers were used in the PCR:IL-6, CCATGCTACATTTGCCGA-3′ and 5′-CTGCGCAGCTTTAAGGAG-3′;IL-8, 5′-CCTTTCCACCCCAAATTTATCA-3′ and 5′-TTTCTGTGTTGGCGCAGTGT-3′;IL-1β, 5′-ACGAATCTCCGACCACCACTA-3′ and 5′-TCCATGGCCACAACAACTGA-3′;TNF-α, TGGCCCAGGCAGTCAGAT-3′ and 5′-GGTTTGCTACAACATGGGCTA-3′;CCL20, 5′-TGTGTATCCAAGACAGCAGTCAAA-3′ and 5′-CCACCTCTGCGGCGAAT-3′;WWOX, 5′-AGTCGCCTCTCTCCAACAAA-3′ and 5′-CGTCTCTTCGCTCTGAGCTT-3′GAPDH, 5’-TGCACCACCAACTGCTTAGC and 5’-GGCATGGACTGTGGTCATGAG.

### 4.7. Enzyme-Linked Immunosorbent Assay (ELISA)

Levels of IL-8, IL-6, TNF-α, and IL-1β secreted into the culture media were determined using commercial ELISA kits; IL-8 (Thermo Scientific, CHC1303), IL-6 (R&D Systems, Minneapolis, MN, USA, DY206), TNF-α (R&D Systems, DY210) and IL-1β (R&D Systems, DY201) and CCL20 (R&D Systems, DY360).

### 4.8. PKC

Cells were harvested and sonicated in lysis buffer (4 mM AEBSF, 1 µg/mL pepstatin, 1 µg/mL leupeptin, 1 µg/mL benzamidine, 1 mM EGTA, 1 mM EDTA, 1 mM sodium fluoride, 1 mM beta-glycerophosphate, 1 mM sodium orthovanadate, and 2.5 mM sodium pyrophosphate). PKC activities in supernatants were assayed using a kit (Abcam, ab139437). Absorbance was measured at 450 nm using a FilterMax F5 Multi-Mode Microplate Reader (Molecular Devices, San Jose, CA, USA).

### 4.9. Statistical Analysis

All data are presented as mean ± standard error of the mean. All statistical analyses were performed using GraphPad Prism software version 9 (GraphPad Software, San Diego, CA, USA). All experiments were repeated at least three times. Student’s *t*-test was used to compare the two groups, and *p*-values of less than 0.05 were considered statistically significant.

## 5. Conclusions

We presented evidence supporting the role of WWOX in modulating the NF-kB pathway in epidermal keratinocytes through calcium and PKC signaling. We demonstrated that WWOX knockdown significantly reduced NF-kB signaling and levels of proinflammatory cytokines, which are essential in psoriatic inflammation. Targeting WWOX could be a novel future therapeutic approach for treating psoriasis. Further studies on the impact of WWOX on the pathogenesis of psoriasis and the potential of targeting WWOX to treat psoriasis are needed.

## Figures and Tables

**Figure 1 ijms-25-00167-f001:**
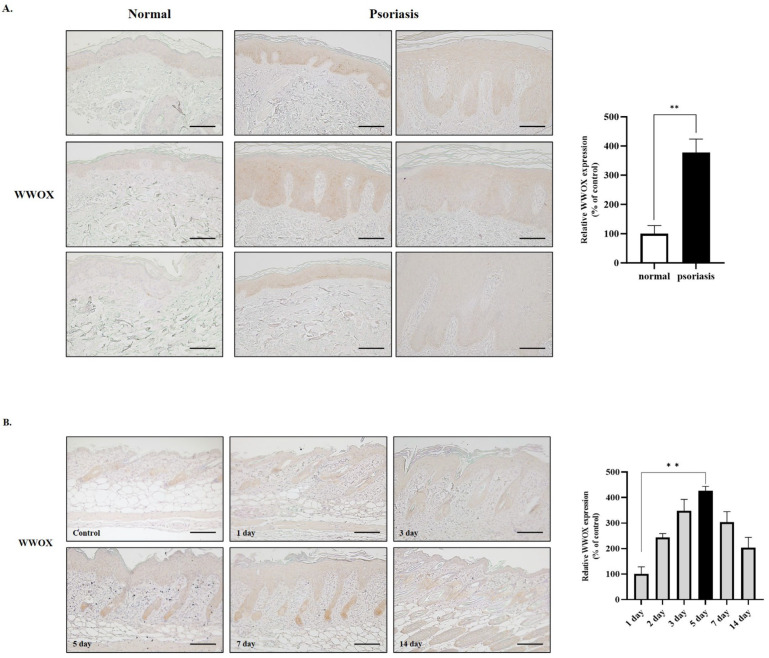
Immunochemical analysis of WW domain-containing oxidoreductase (WWOX) expression in the psoriatic skin and imiquimod (IMQ) mouse model. (**A**) Paraffin-embedded normal skin and psoriatic lesions were immunohistochemically stained with WWOX antibody. The WWOX level in psoriatic lesions was higher than that of the normal epidermis (normal = 3, psoriasis patient = 6, Magnification: 200×, Scale bar = 100 μm). (**B**) Skin sections from IMQ mice were immunohistochemically stained with WWOX antibody. WWOX level in the imiquimod-treated epidermis was increased compared to that in the control epidermis (*n* = 3, Magnification: 200×, Scale bar = 100 μm). The histogram shows statistical results for the immunohistochemical evaluation for WWOX in epidermis. All data are presented as mean ± SEM. Student’s *t*-test was performed for statistical analysis. ** *p* < 0.01.

**Figure 2 ijms-25-00167-f002:**
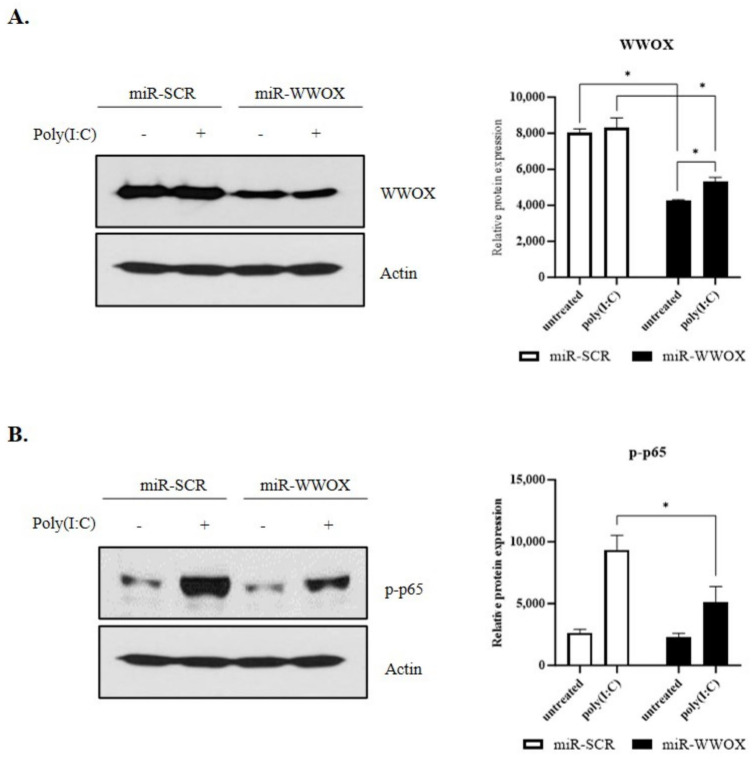
Effect of WW domain−containing oxidoreductase (WWOX) knockdown on nuclear factor kappa B (NF-kB) signaling. (**A**) WWOX protein expression in epidermal keratinocytes was verified by Western blot. (**B**) Poly(I:C)-induced protein expression of NF-kB marker phospho-p65 was decreased by WWOX knockdown. Histogram of relative protein expression. Data are presented as mean ± SEM (*n* = 3). Student’s *t*-test was performed for statistical analysis. * *p* < 0.05.

**Figure 3 ijms-25-00167-f003:**
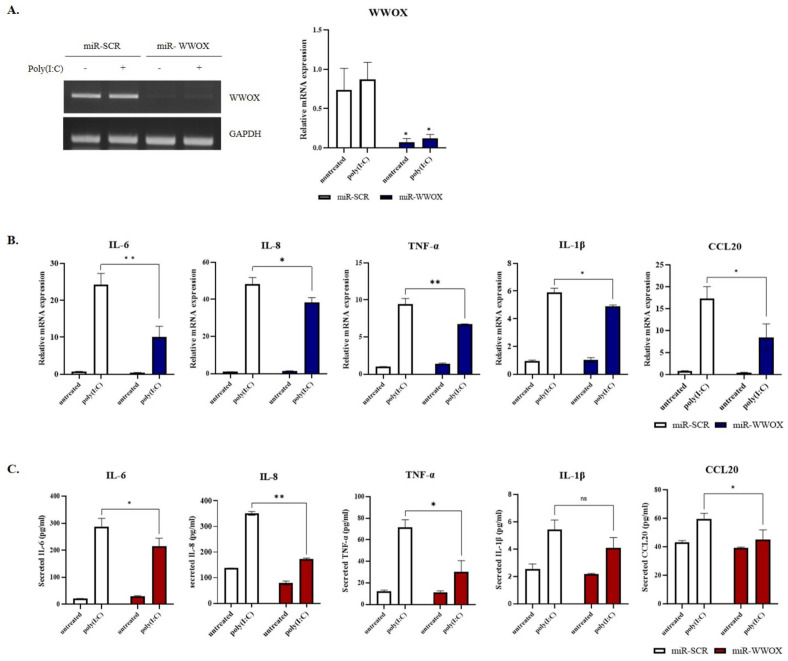
Effects of WW domain-containing oxidoreductase (WWOX) knockdown on poly(I:C)−induced inflammatory reaction in keratinocytes. (**A**) WWOX gene expression in epidermal keratinocytes was verified by real-time polymerase chain reaction. (**B**) Poly(I:C)-induced messenger RNA expression levels of Interleukin (IL)-6, IL-8, tissue necrosis factor-alpha (TNF-α), IL-1β, and chemokine ligand (CCL20) were decreased by WWOX knockdown. (**C**) The conditioned media were collected, and secretion levels of IL-6, IL-8, TNF-α, and IL-1β were determined by enzyme-linked immunosorbent assay. WWOX knockdown inhibited poly(I:C)-induced secretion of inflammatory cytokines. Data are presented as mean ± standard error (*n* = 3). Student’s *t*-test was performed for statistical analysis. * *p* < 0.05, ** *p* < 0.01.

**Figure 4 ijms-25-00167-f004:**
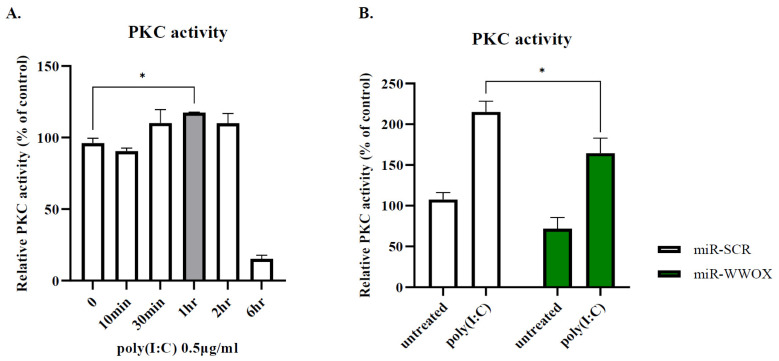
PKC activity is decreased after WW domain-containing oxidoreductase (WWOX) knockdown. (**A**) Epidermal keratinocytes were treated poly(I:C) in a time-dependent manner. The largest increase was found at 1 h. (**B**) Poly(I:C)-induced PKC activity was decreased by WWOX knockdown. Data are presented as mean ± SEM (*n* = 3). Student’s *t*-test was performed for statistical analysis. * *p* < 0.05.

**Figure 5 ijms-25-00167-f005:**
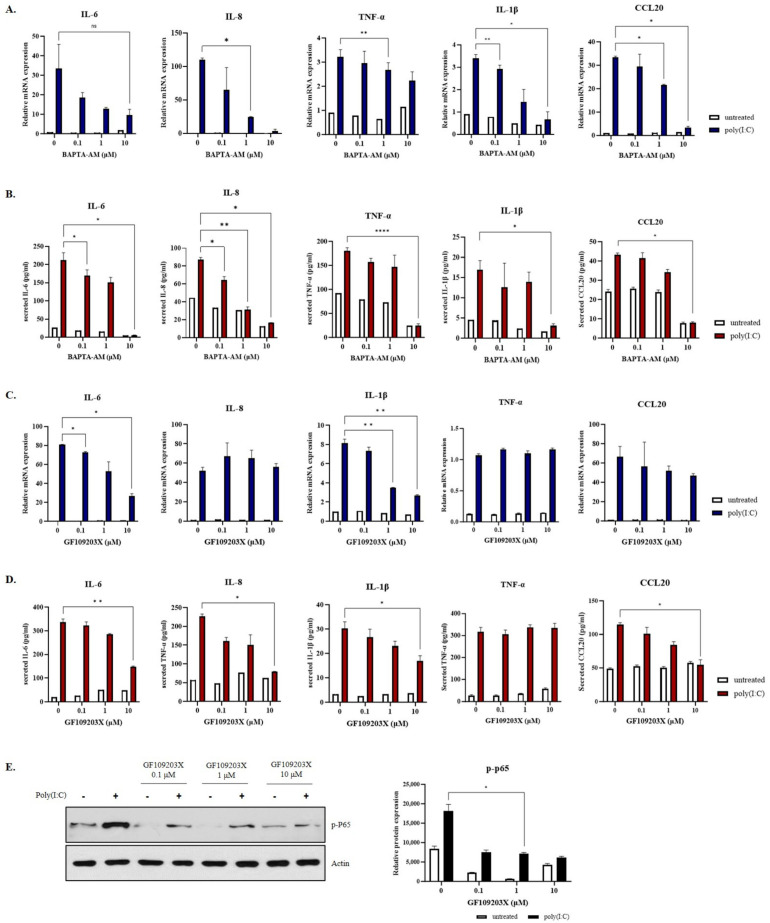
Effects of Ca^2+^ and protein kinase C (PKC) on the inflammatory response. (**A**,**B**) Epidermal keratinocytes were pretreated with BAPTA−AM and then treated with poly(I:C). BAPTA-AM reduced the messenger RNA (mRNA) expression of interleukin (IL)-6, IL-8, tissue necrosis factor-alpha (TNF-α), IL-1β, and chemokine ligand (CCL20) in a dose-dependent manner. Secretion levels of IL-6, IL-8, TNF-α, and IL-1β were also decreased gradually in a dose-dependent manner. (**C**) Epidermal keratinocytes were pretreated with GF109203X at different doses and then treated with poly(I:C). GF109203X reduced the mRNA expression of IL-6, IL-8, IL-1β, and CCL20. (**D**) Secretion levels of IL-6, IL-8, IL-1β, and CCL20 were decreased gradually by dose. (**E**) GF109203X also significantly reduced the protein expression of nuclear factor kappa B (NF-kB) marker phospho-p65 by Western blot. Data are presented as mean ± SEM (*n* = 3). Student’s *t*-test was performed for statistical analysis. * *p* < 0.05, ** *p* < 0.01, **** *p* < 0.0001.

## Data Availability

The data used to support the results of this study are available from the corresponding authors upon request.

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
