# Peer review of "Mechanistic Investigation of WWOX Function in NF-kB-Induced Skin Inflammation in Psoriasis"

_ijms, 2023, doi:10.3390/ijms25010167_

Round 1

Reviewer 1 Report (Previous Reviewer 3)

Comments and Suggestions for Authors

The authors responded the comment about non specific targeting of a miRNA by conducting a rescue experiment. However, in cells with Poly IC treatment, the association of WWOX and p-p65 is weak. In miR-scr-polyIC, p-p65 was higher than in miR-wwox+ad/ha-wwox in supplement figure 4. However, the authors proposed that WWOX positively regulated p-p65. Please explain this contradition.

Author Response

The authors responded the comment about non specific targeting of a miRNA by conducting a rescue experiment. However, in cells with Poly IC treatment, the association of WWOX and p-p65 is weak. In miR-scr-polyIC, p-p65 was higher than in miR-wwox+ad/ha-wwox in supplement figure 4. However, the authors proposed that WWOX positively regulated p-p65. Please explain this contradiction.

Reply: Thank you for your thorough review and valuable insights on our manuscript. Following your suggestion, we performed a rescue experiment which revealed that WWOX overexpression significantly increased the levels of p-p65, NF-kB marker, thereby reversing the reduction caused by WWOX knockdown. We have incorporated these findings as Supplementary Figure 4 in the revised manuscript, bolstering the validity of our results. Regarding the observed lower p-p65 level in the group where WWOX was rescued after knockdown with microRNA compared to the microRNA-scrambled group, the reason remains uncertain. One possibility is that the WWOX knockdown via miR-WWOX was so effective that, although the subsequent rescue experiment with overexpression showed WWOX’s influence on p-p65 levels, the recovery didn’t quite reach the pre-knockdown levels. It is also conceivable that other factors may have influenced these results. We recognize the need for further detailed investigation to fully understand these mechanisms. We have included this consideration as a potential limitation in the revised version of our manuscript.

Reviewer 2 Report (Previous Reviewer 1)

Comments and Suggestions for Authors

The manuscript entitled "Mechanistic Investigation of WWOX Function in NF-kB-induced Skin Inflammation in Psoriasis" by Min Jeong Shin et al., was presented as a Original Article in IJMS, into the Section "Molecular Pathology, Diagnostics, and Therapeutics" - Special Issue: Challenges and Future Trends of Inflammatory Skin Diseases Treatment. The reviewer has already revised the manuscript before that it was withdrawn after the first revision round. She noticed that some points of the her previous revisions were resolved, however the reviewer has still some concerns:

1. The authors consider the possibility of analyzing the expression of WWOX in a human array and/or expression profiling by high throughput sequencing available in GEO DataSets

2. What is the reason why the authors used female mice in the IMQ-induced psoriasis mouse model? Is there a specific reason behind the sex choice (i.e. hormonal influence)? Please, explain

3. It is questionable that the authors used a single microRNA sequence to target human WWOX messenger RNA

Author Response

The manuscript entitled “Mechanistic Investigation of WWOX Function in NF-kB-induced Skin inflammation in Psoriasis” by Min Jeong Shin et al., was presented as a Original Article in IJMS, into the Section “Molecular Pathology, Diagnostics, and Therapeutics” – Special Issue: Challenges and Future Trends of Inflammatory Skin Diseases Treatment. The reviewer has already revised the manuscript before that it was withdrawn after the first revision round. She noticed that some points of the her previous revisions were resolved, however the reviewer has still some concerns:

  1. The authors consider the possibility of analyzing the expression of WWOX in a human array and/or expression profiling by high throughput sequencing available in GEO DataSets.

Reply: We appreciate the reviewer for highlighting this aspect. We concur with the reviewer on the importance of examining WWOX expression through human arrays like gene chips or tissue arrays. However, due to constraints in the revision timeframe and challenges such as obtaining sufficient skin biopsies, it is currently difficult for us to carry out such experiments. Therefore, we resorted to analyzing WWOX expression data available through the Gene Expression Omnibus (GEO) Profiles provided by NCBI. As a result, we successfully located public data investigating WWOX expression in both lesional and non-lesional skin biopsies from psoriasis patients (https://www.ncbi.nlm.nih.gov/geoprofiles?term=GDS5392[ACCN]+WWOX). This dataset involved skin biopsies from four donors, comparing gene expression patterns using the Affymetrix U133 Human Genome Plus 2.0 arrays. Among the 5 WWOX probe sets included in the Affymetrix U133 Human Genome Plus 2.0 arrays, we focused on the probe set 221147_x_at, with its data presented in Supplementary Figure 2. Notably, in this public dataset, WWOX expression was higher in lesional skin compared to non-lesional skin, aligning with our results (Supplementary Figure 2A).
We have briefly included a description of this analysis in the revised manuscript.

  1. What is the reason why the authors used female mice in the IMQ-induced psoriasis mouse model? Is there a specific reason behind the sex choice (i.e. hormonal influence) ? Please, explain.

Reply: Thank you for your insightful comment. We chose female mice for our imiquimod-induced psoriasis model due to their more consistent hair growth cycle compared to male mice. Specifically, we utilized female mice aged 7~8 weeks, aligning with the telogen phase of their hair growth cycle. Male mice are generally less preferred for such experiments due to their less synchronized hair growth cycles and the higher variability compared to females. Additionally, male mice housed together often engage in aggressive behavior, leading to back skin wounds and inflammation. Therefore, female mice were deemed more appropriate for our study’s objectives.

  1. It is questionable that the authors used a single microRNA sequence to target human WWOX messenger RNA.

Reply: Thank you for your valuable feedback. In response to your comment, we have included additional details in our revised manuscript regarding the microRNA-expressing adenovirus used to target WWOX. The specific microRNA sequence employed in our study is ggcgtttactgtggatgataa, corresponding to nucleotides 395 to 415 in NM_016373.4. This sequence was chosen based on previous research (Kwak et al, Exp Dermatol 2015;24:942-946) demonstrating its efficacy in inhibiting WWOX expression. In the current study, we utilized an adenovirus targeting this exact microRNA sequence to explore its effects on WWOX. We believe that providing these details enhances the clarity and replicability of our work, and we appreciate the opportunity to strengthen our manuscript with this additional information.

Reviewer 3 Report (Previous Reviewer 2)

Comments and Suggestions for Authors

Article is improved after revisions 

Suitable for publication 

Author Response

Article is improved after revisions. Suitable for publication.

Reply: We really appreciate the reviewer for the positive feedback. We appreciate your valuable feedback, which has been instrumental in refining our manuscript. Thank you once again for your guidance and support in this process.

Round 2

Reviewer 1 Report (Previous Reviewer 3)

Comments and Suggestions for Authors

no further concern

Author Response

no further concern

Reply: We are genuinely grateful to the reviewer for the positive feedback. Your valuable input has played a pivotal role in improving our manuscript. Once more, we extend our heartfelt thanks for your guidance and unwavering support throughout this process.

Reviewer 2 Report (Previous Reviewer 1)

Comments and Suggestions for Authors

The reviewer approves the revised version of the manuscript but has one last request. It would be appropriate to show representative images of back skin of IMQ induced-psoriasis experiment and add some characterization and evaluation of optimized IMQ-induced psoriasis model, such as representative H&E staining and critical parameters like the skin thickness changes. 

Author Response

The reviewer approves the revised version of the manuscript but has one last request. It would be appropriate to show representative images of back skin of IMQ induced-psoriasis experiment and add some characterization and evaluation of optimized IMQ-induced psoriasis model, such as representative H&E staining and critical parameters like the skin thickness changes.

Reply: Thank you for your constructive feedback and guidance. In response to your suggestion, we have included representative images and detailed findings of Hematoxylin and Eosin (H&E) staining at different time points from the IMQ-induced psoriasis mouse experiment. We have also provided a comparative analysis of epidermal thickness between the IMQ-induced and control groups, specifically focusing on day 5, which marks the peak of psoriasis induction. This analysis and corresponding results have been illustrated in Supplementary Figure 1. These additions provide a more comprehensive understanding of the effects at the critical point of psoriasis induction and enhance the overall quality of our manuscript. Thank you for your invaluable input in making these important improvements.

This manuscript is a resubmission of an earlier submission. The following is a list of the peer review reports and author responses from that submission.

Round 1

Reviewer 1 Report

Comments and Suggestions for Authors

The manuscript entitled "Mechanistic Investigation of WWOX Function in NF-kB-induced Skin Inflammation in Psoriasis" by Min Jeong Shin et al., was presented as a Original Article in which the authors investigated the association of WWOX with psoriatic inflammation, highlighting the possibility of targeting WWOX as a novel therapeutic approach for psoriasis treatment. The manuscript, presented into the Section "Molecular Pathology, Diagnostics, and Therapeutics" - Special Issue: Challenges and Future Trends of Inflammatory Skin Diseases Treatment, could be of potential interest for a wide community working on Skin Pathologies, arousing interest not only for basic researchers, but also for clinicians. The manuscript is fine organized in terms of data presentation and writing, resulting accessible and complete to the reader. The Introduction section is fine described, and the paper appears to be well-designed. However, the reviewer has some concerns:

1. The authors consider the possibility of analyzing the expression of WWOX in a human array

2. What is the reason why the authors used male mice in the IMQ-induced psoriasis mouse model? Is there a specific reason behind the sex choice (i.e. hormonal influence)?

3. The reviewer suggests to show a schematic graph summarizing a timeline (start and end/sacrifice) for the treatments used in the IMQ-induced psoriasis mouse model

4. It is questionable that the authors used a single microRNA sequence to target human WWOX messenger RNA

5. The Materials and methods section could be further improved providing information about all reagents and equipment used (manufacturers/producers and code/catalog number)

6. In the original IHC analysis, scale bars are not visible; make them more evident and specify the magnification (x) and the length (μm) in the Figure Legend.

7. No quantification of Western Blot analyses is shown. Please, provide quantification histograms (ratio of target protein levels relative to ACTIN/GAPDH levels) of all Western Blot analyses to determine changes in target proteins expression. Please, provide the original uncropped and unprocessed scans of the blots in the section of Original Images for Blots/Gels. This should be cited once in the Materials and Methods section (2.5. Western blot analysis). 

Reviewer 2 Report

Comments and Suggestions for Authors

Very interesting and well-written article 

The authors showed that deletion of WWOX significantly reduced NF-kB signaling and levels of proinflammatory cytokines, which are essential in psoriatic inflammation.

I only have minor revisions for the authors 

- 1) In the introduction, new drugs for psoriasis are mentioned but many references of real life and new drugs to date are missing, I leave some references for authors to use

- DOI: 10.2147/PTT.S407647

- DOI: 10.2147/TCRM.S388324

- DOI: 10.1007/s40265-020-01261-8

2) Minimal language revision is required

3) Some typos are present so the whole text needs to be revised

4) Add paragraph limitations of the study

5) Materials and methods need to be improved in terms of accuracy and extrapolation of data

Comments on the Quality of English Language

Minor editing of English language required

Reviewer 3 Report

Comments and Suggestions for Authors

In this manuscript, the authors investigated the role of WWOX in psoriasis. They showed that WWOX was upregulated in lesional skins and downregulation of WWOX attenuated polyIC induced cytokine and chemokine expression. However, several major concerns need to be addressed.

  1. The authors used uncommon approach: a microRNA targeting WWOX to inhibit WWOX expression. Compared to siRNA, a microRNA usually has many targets. This raises a concern about the specificity of such knockdown. It's possible that many other genes were knockdown by this microRNA. If so, it can't conclude that the observations in this manuscript is due to WWOX. In addition, the efficiency of knockdown by this microRNA seems low. One way to address this issue is to conduct a rescue experiment by overexpression of WWOX in cells transduced with microRNA against WWOX.
  2. Please describe detail about this microRNA for WWOX. Sequence?
  3. It's confusing to use polyIC to induce inflammatory response. In this manuscript, the authors attempted to examine the role of WWOX in psoriasis. However, polyIC usually is used for mimic infection-induced inflammation but not for psoriasis.
  4. A densitometry and quantification analysis need to be done for western blots.
  5. The authors need to provide justification why they investigated PKC signaling.
  6. It has been shown that WWOX was upregulated in lesional psoriatic skins. So the relationship between WWOX and psoriasis was investigated.

Deepti Verma 1 2, Anna-Karin Ekman 1 2, Cecilia Bivik Eding 1, Charlotta Enerbäck 1.Genome-Wide DNA Methylation Profiling Identifies Differential Methylation in Uninvolved Psoriatic Epidermis.

Journal of Investigative Dermatology

Volume 138, Issue 5, May 2018, Pages 1088-1093

Suarez-Farinas et al., 2012. Expanding the psoriasis disease profile: interrogation of the skin and serum of patients with moderate-to-severe psoriasis

J Invest Dermatol, 132 (2012), pp. 2552-2564